# Simulations of scenarios for urban household water and energy consumption

**Marco Casazza**[1,2]*, **Jingyan Xue**[3], **Shupan Du**[3], **Gengyuan Liu**[3,4]*, **Sergio Ulgiati**[1,3,4]

**1** Department of Sciences and Technologies, University of Napoli 'Parthenope', Centro Direzionale, Napoli, Italy, **2** Interdepartmental Research Centre on Urban and Event Studies (OMERO), University of Torino, Torino, Italy, **3** State Key Joint Laboratory of Environment Simulation and Pollution Control, School of Environment, Beijing Normal University, Beijing, China, **4** Beijing Engineering Research Center for Watershed Environmental Restoration & Integrated Ecological Regulation, Beijing, China

* marco.casazza@uniparthenope.it (MC); liugengyuan@bnu.edu.cn (GL)

## Abstract

The expansion of cities and their impacts currently constitutes a challenge for the achievement of sustainable development goals (SDGs). In this respect, assessments of resource consumption and the delivery of appropriate policies to support resource conservation are of paramount importance. Previous works in the literature have focused on one specific resource (e.g., water, energy, food) at the household level, while others have analysed the inter-relations among different resources (i.e., the nexus approach) at larger spatial scales (e.g., urban level). Moreover, household behavioural attitudes are generally excluded while assessing resource consumption scenarios. This work overcomes previous limitations by proposing a causal-loop structure derived from the literature, from which simulations of different scenarios can be generated that consider the nexus between food, energy and water at the household level. These simulations can provide alternative scenarios to assess the impacts of monetary policies as well as education and communication actions on the enhancement of resource savings and consider both their current use and household preferences. The metropolitan area of Napoli was chosen as the testbed area for the simulations. The results, in relation to the testbed, proved that communication actions would be most appropriate to increase the level of resource savings. The business-as-usual scenario was especially sensitive to variations in individual preferences towards pro-environmental behaviours and showed their higher impacts on the results. Improvements of this method and its derived scenarios in the context of the urban planning process could support the implementation of informed policies towards the conservation of key resources and promotion of sustainable citizen behaviour.

**Data Availability Statement:** All relevant data are within the manuscript and its Supporting Information files.

**Funding:** The authors gratefully acknowledge the support from the Italian Ministry of Foreign Affairs and International Cooperation (Ministero degli

## Introduction

Energy poverty and water and food scarcity are key obstacles to the achievement of sustainable development goals (SDGs) [1]. Currently, 900 million people do not have access to electricity, and 2.1 billion people rely on biomass burning for cooking [2]. Such a lack of access depends

Affari Esteri e della Cooperazione Internazionale - Direzione Generale per la Promozione del Sistema Paese; Grant number: PGR05278) and National Natural Science Foundation of China (No. 52070021). The authors also gratefully acknowledge the support from the projects "Realizing the Transition towards the Circular Economy: Models, Methods and Applications (ReTraCE)", funded by the H2020-MSCA-ITN-2018 programme (Grant Number: 814247) and "Promoting Circular Economy in the Food Supply Chain (ProCEedS)", funded by the H2020-MSCA-RISE-2018 programme (Grant Number: 823967). The funders had no role in study design, data collection and analysis, decision to publish, or preparation of the manuscript.

**Competing interests:** The authors have declared that no competing interests exist.

on several factors, among which energy prices, low incomes and energy efficiency are considered to be the most relevant [3]. Globally, four billion people face problems of severe water scarcity [4]. According to the Food and Agriculture Organization of the United Nations (FAO), 690 million people suffered from undernourishment in 2019 [5]. By analysing these challenges at the household scale, subjective factors become relevant in coping with resource poverty and in adopting sustainable strategies to improve population livelihoods. The current COVID-19 pandemic has further exacerbated the existing survival uncertainty and has generated an increase in anxiety responses [6].

Appropriate and adaptive strategies are necessary to design the necessary roadmaps that will eradicate poverty and are in agreement with the SDGs. Due to the complex nature of consumption systems and the connections among different resource flows, a holistic approach is necessary to recognize the existing mutual relationships within the life cycles of different resources. Moreover, policies and roadmaps should be supported by appropriate analytical tools. Several studies have recognized and modelled the connections among different resources. This is the case for studies of the so-called food-energy-water (FEW) nexus [7–9]. Fewer studies have focused on dynamic simulations of such systems and their ontology [10–13], and these studies have had some limitations. The first limitation is related to the system scale studied. In particular, the available studies have often been limited to the urban scale or, if developed at the household scale, they have focused on one resource (e.g., water). The second limitation is related to the details that are included in the simulations. In particular, individual preferences, being relevant for the adoption of more sustainable solutions, were often excluded from the simulations.

This study aims to overcome the previous limitations and proposes a system dynamic analysis of food, energy and water consumption at the household level. In particular, the purpose of the work is to identify the structural features and evolution of the FEW consumption system at the household level in an urban context. Definition of such systemic properties can support the visualization of alternative scenarios that can support policy makers in defining tailored solutions to manage the sustainable demand of such resources and to alleviate the current state of resource poverty, which affects many people worldwide [14]. Furthermore, end user preferences were included in the analysis to determine their impacts on the consumption of key resources that were considered in this research. In the following section, the model structure and scenarios considered are described, in addition to the details that are related to the scenario simulator testbed. The results of the application of the method, in the form of different scenarios, are detailed in the results section. The sensitivity of the business-as-usual (BAU) scenario with respect to household behavioural preferences will be discussed to identify the variations in derived resource consumption data on behavioural parameters that are necessary as inputs to the simulations. Then, the potential use and applications of this method as a tool for urban planning are discussed in light of supporting a transition towards the conservation of key resources and a higher adoption rate of pro-environmental behaviours. On this basis, the conclusions are given in the final section.

## Materials and methods

### Household FEW nexus calculator components

The basic model for the proposed household FEW consumption simulator was based on the system components that were identified in a previous study [15]. The model was subdivided into four interfaces that were related to energy, water and food consumption, plus a combined "FEW nexus" interface. The selected system factors included the elements that affected daily

resource consumption, such as residents' habits and preferences, as well as the characteristics and preferences that are related to water and energy technologies.

Energy subsystem end uses included two parts: cooking and noncooking. Noncooking end uses included water heating, laundry, lighting, heating and cooling, electronic entertainment equipment and refrigeration. Water consumption was disaggregated in the same manner as energy. Noncooking water uses included washing, laundry, toilet flushing and showering. The water devices considered included faucets, washing machines, toilets and showers. Seven types of energy or water appliances were considered, including showers, washing machines, lamps, air conditioners, TVs, computers and refrigerators.

Total water and the total energy amounts were obtained by summing all water and energy end uses. For both water and energy subsystems, the factors which directly influence their consumption are appliance ownership, frequency and duration of each end use and efficiency of each use. Furthermore, an additional factor was the rate of renewable energy utilization. Both appliance efficiency and efficiency in terms of duration were considered. Appliance efficiency is related to the quality standards of the devices, which reflect whether the equipment is efficient or not. Being different from appliance efficiency, duration efficiency is influenced by behavioural measures in terms of educational campaigns that involve key stakeholders and citizens.

In terms of the food subsystem, we applied two broad classifications: "meat & dairy" and "nonmeat & nondairy" diets. Changes in dietary patterns were reflected by the percentages of meat and dairy or nonmeat and nondairy food consumption. This broad subdivision depends on the higher impact of meat and dairy diets on greenhouse gas emissions when compared to other diets [16]. Additional dietary specifications were excluded from the current version of this model. Further details about the simulator structure and subcomponents are given in the Supplementary Material (S1 File).

## Scenario design

The proposed household FEW nexus model is used to explore the influence of residents' daily behaviours and practices on household end uses. In particular, a long-term dynamic model is proposed that is capable of simulating alternative energy, gas and water consumption scenarios from 2010 to 2050. Based on the key factors that may determine household FEW consumption, various behavioural and policy scenarios are considered to explore their effectiveness. Middle-term (until 2035) and long-term (until 2050) timeframes are included in the study.

As shown in Tables 1–3, thirty-four policy scenarios, which are potentially able to reduce household consumption or household $CO_2$-eq emissions, are analysed in this research. All of these scenarios are divided into 2 main categories: price strategies (P) and nonprice strategies (B). Price increases are recognized as effective policy tools for urban resource conservation. Households may choose some resource-saving measures to reduce their daily costs that are caused by increased unit prices of resources. Four levels, which are based on the current price of each resource type, are simulated here to evaluate the household resource efficiency and climate change potential in Naples if the prices of electricity, gas, water and meat and dairy foods increase by 40%, 60%, 80%, and 100%, respectively.

In contrast to price strategies, nonprice strategies are focused on adjustments of household daily behaviours (B). In particular, they are related to information and education. Two kinds of knowledge, including behavioural guidance and household enhancement measures (BGs) and general education and public adjustment measures (BEs), are considered. The specific behavioural guidance and household enhancement measures concentrated on providing specific information guidance regarding saving behaviours to encourage residents to perform

**Table 1. Overview of FEW conservation strategies—price strategies (P).**

| No. | Scenario description |
|---|---|
| P1 | Increase average gas price rate by 40%. |
| P2 | Increase average gas price rate by 60%. |
| P3 | Increase average gas price rate by 80%. |
| P4 | Increase average gas price rate by 100%. |
| P5 | Increase average electricity price rate by 40%. |
| P6 | Increase average electricity price rate by 60%. |
| P7 | Increase average electricity price rate by 80%. |
| P8 | Increase average electricity price rate by 100%. |
| P9 | Increase average water price rate by 40%. |
| P10 | Increase average water price rate by 60%. |
| P11 | Increase average water price rate by 80%. |
| P12 | Increase average water price rate by 100%. |
| P13 | Increase average meat and dairy price rate by 40%. |
| P14 | Increase average meat and dairy price rate by 60%. |
| P15 | Increase average meat and dairy price rate by 80%. |
| P16 | Increase average meat and dairy price rate by 100%. |

more intensive sustainable behaviours, such as reducing shower frequency, reducing shower length or choosing more plant-based foods. However, general education and public adjustment measures concern the availability of households perceptions regarding the environmental and the long-term benefits of implementing sustainable behaviours or by retrofitting efficient appliances by providing general information through environmental communication campaigns (e.g., newspapers, TV advertisements, and posters). Details of the simulation parameters are given in the Supplementary Material (S2 in S1 File).

## Scenario testbed

The proposed model was applied to the metropolitan city of Naples (southern Italy). Naples is the third largest city in Italy. The province-level municipality (i.e., the metropolitan city) population consists of 3 million inhabitants and has a high population density and high level of urbanization. The number of households recorded by ISTAT in 2019 was 1,132,408.

The urban area is characterized by great contradictions, both in economic and social terms, as leading-edge realities coexist with conventional realities. Situations of extreme energy

**Table 2. Overview of FEW conservation strategies—general education & public will adjustment measures (BE).**

| No. | Scenario description |
|---|---|
| BE1 | Provide more knowledge of adopting resource-saving appliances through newspaper, TV advertisements, and posters. (60% increase in implementation strength) |
| BE2 | Provide more knowledge of implementing sustainable behaviours and practices through newspaper, TV advertisements, and posters. (80% increase in implementation strength) |
| BE3 | Provide more knowledge of implementing sustainable behaviours and practices through newspaper, TV advertisements, and posters. (100% increase in implementation strength) |
| BE4 | Provide more knowledge of implementing sustainable behaviours and practices through newspaper, TV advertisements, and posters. (60% increase in implementation strength). |
| BE5 | Provide more knowledge of implementing sustainable behaviours and practices through newspaper, TV advertisements, and posters. (80% increase in implementation strength). |
| BE6 | Provide more knowledge of implementing sustainable behaviours and practices through newspaper, TV advertisements, and posters. (100% increase in implementation strength). |

**Table 3. Overview of FEW conservation strategies—specific behavioural guidance & household enhancement measures (BG).**

| No. | Strategy description |
|---|---|
| BG1 | Reduce the average daily washing length by 20% each time. |
| BG2 | Reduce the average daily washing length by 40% each time. |
| BG3 | Reduce the average daily washing length by 60% each time. |
| BG4 | Reduce the shower frequency by 20%. The average shower frequency will be 0.46/cap/day. |
| BG5 | Reduce the shower frequency by 40%. The average shower frequency will be 0.34/cap/day. |
| BG6 | Reduce the shower frequency by 60%. The average shower frequency will be 0.23/cap/day. |
| BG7 | Reduce the fraction of meat & dairy consumption by approximately 5% on the current basis. The average fraction of meat & dairy consumption will be 0.3. |
| BG8 | Reduce the fraction of meat & dairy consumption by approximately 10% on the current basis. The average fraction of meat & dairy consumption will be 0.25. |
| BG9 | Reduce the fraction of meat & dairy consumption by approximately 15% on the current basis. The average fraction of meat & dairy consumption will be 0.2. |
| BG10 | Preserve the resource curtailment habits to 2 years. |
| BG11 | Preserve the maintenance of resource curtailment habits to 3 years. |
| BG12 | Preserve the maintenance of resource curtailment habits to 4 years. |

efficiency coexist with others of absolute poverty. Overall, per capita GDP is low, and the real unemployment rate is especially high, with a strong hidden economy. Therefore, it is necessary to find appropriate solutions to target the existing social disparities. Mild climate conditions do not prevent the region from suffering from poverty, which depends mainly on the economic and social factors described above. In particular, low average per capita GDP, high unemployment rates, the use of old and inefficient appliances and devices, poor education, high environmental pollution from energy sources and a large share of old and popular residential homes are among the main causes of the current situation. It is likely that this area, which already suffered from the effects of the 2008 economic crisis, will also be impacted by the COVID-19 pandemic crisis. Under these circumstances, many households have considerably reduced their energy expenditures [17]. Finally, different citizen behaviours may also affect the possibility of defining a clear scenario of household resource poverty [18]. In fact, behavioural patterns can hinder the social recognition of such poverty while causing negative consequences on the well-being and health of the population [19].

## Data collection

Data were collected from official statistical publications and studies as well as from a field survey. Population data for the metropolitan city of Naples are freely available from the website of the Italian National Institute of Statistics in the section that is related to population and household (ISTAT: www.istat.it/en/population-and-households). Demographic projections are also available from the ISTAT website (www.istat.it/it/files/2018/05/previsioni_demografiche.pdf). Fig 1 shows the real (from 2010 to 2019) and projected (from 2020 to 2050) population trends according to ISTAT official data and projections.

The preliminary determinants and characteristics, including user preferences regarding energy [20–22] and water demand [23–26] in Italy, were derived from the scientific literature. The latest details for household energy consumption were derived from an official report that was published by the Italian Agency for Energy Efficiency (ENEA) [27]. Further details, including updated energy and gas prices, were derived from another report that was published

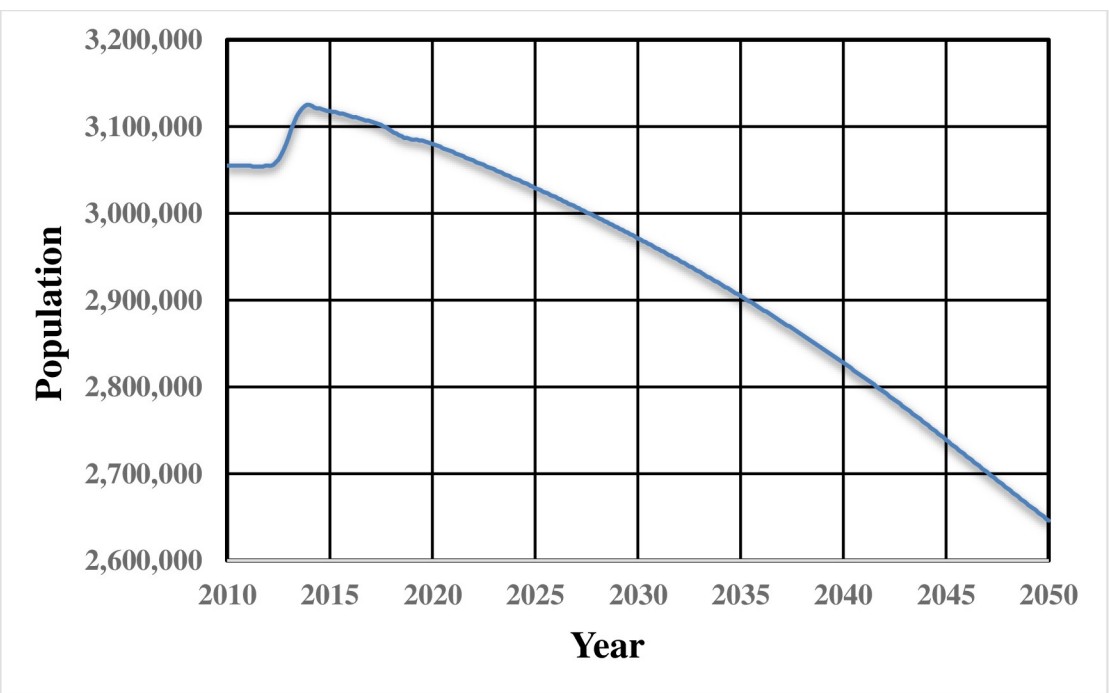

**Fig 1. Actual Naples metropolitan city population data (years 2010–2019) and projected data (years 2020–2050) (data source: Italian National Institute of Statistics–ISTAT).**

by ENEA [28]. Data for household water consumption were reported on the ISTAT website and in yearly reports on the quality of the Italian environment that were published by the Italian Institute for Environmental Protection and Research (ISPRA).

To further tune the data to the study area, a survey was submitted to 248 university students who were living in the metropolitan city of Naples and functioned as volunteers to serve as additional test subjects. The questionnaire used collected data regarding the number of people living in the same house, water and energy consumption, habits, behavioural and technological preferences, and monthly energy and water bills. This data collection procedure fulfilled the requirements of informed consent [29]: respect for persons, beneficence, and justice. In particular, individual autonomy was considered. No minors or other people with diminished autonomy were involved in the study. The purpose and details of data use were disclosed prior to data collection, which led to written approval for participating in the survey prior to data collection. All human subjects were treated ethically, and their privacy was respected, which are in agreement with the European General Data Protection Regulation [30]. The questionnaires were completely anonymous, and no sensible data (e.g., clinical data, data enabling the direct or indirect identification of participants, and data about individual beliefs or orientations) or data that would enable the subsequent identification of participants were collected during the survey. Finally, the choice of participation in or abstention from the survey would not generate any type of subsequent benefit or harm to the participants. For such reasons, due to the total anonymity involved, the involvement of volunteers, the characteristics of the questionnaire and informed consent, according to the current regulations, no further approval from the internal institutional review board (ethics committee) of the university was necessary. The results of the data collection are summarized in the Supplementary Material (S3 in S1 File).

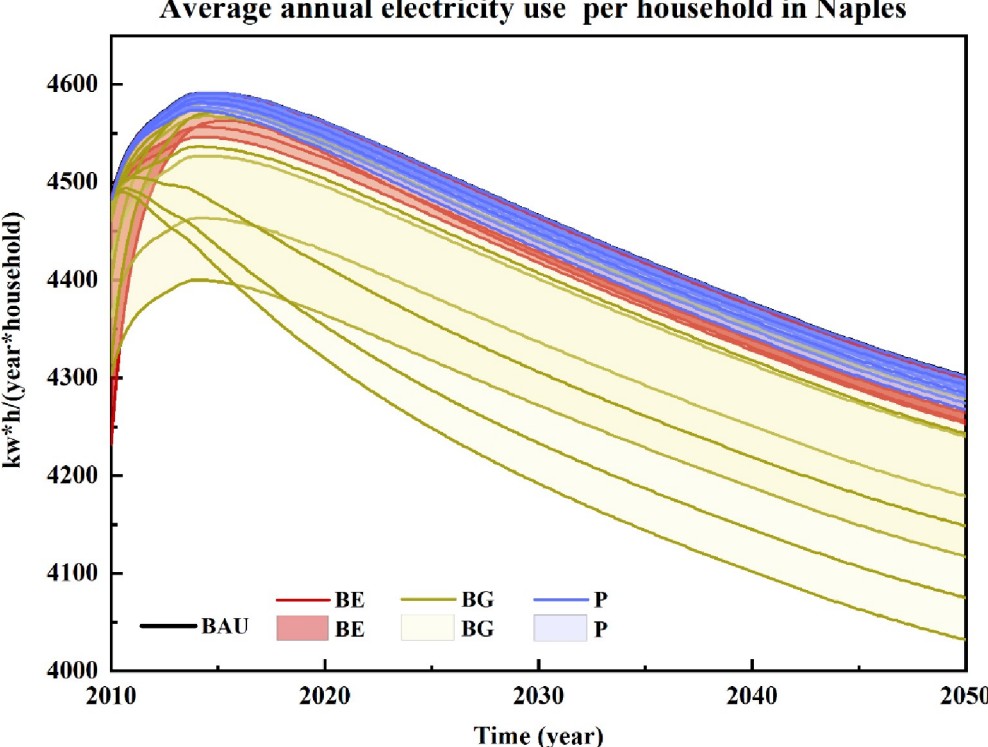

**Fig 2. Scenarios of average annual household electricity use in the metropolitan city of Naples for the years 2010–2050.** Included scenarios: Business as usual (BAU), Price-based policies (P), Behavioural education actions (BE), and Behavioural guidance actions (BG).

## Results

Detailed results of the percentage variations of electricity, gas and water use as a function of the different scenarios with respect to the BAU scenario are reported in the Supplementary Material (S4 in S1 File).

### Average annual household electricity use

Fig 2 shows the scenarios for average annual household electricity use in the metropolitan city of Naples. Educational actions (BE) will not significantly influence household electricity consumption in Naples. In this respect, electricity consumption varied from 0 to 1.35% (3 to 59 kWh/household/year). The perception of pro-environmental behaviour relevance (BE4 to BE6) was higher than that related to resource-saving appliances. The difference was motivated by the additional investment needed to purchase the devices. Behavioural guidance had the best regulation effect on electricity consumption and was independent of the time horizon considered for the scenarios.

In detail, household electricity savings could grow in combination with a reduction in the time spent using water for personal hygiene (BG1 to BG3). The trends are partially motivated by the expected population decrease in the area. According to our simulation results, household electricity savings could reach 66 kWh/household/year (3.2 minutes), 132 kWh/household/year (2.4 minutes) and 198 kWh/household/year (1.6 minutes), respectively. It is worth mentioning that the policy scenarios that focused on increasing the maintenance of resource curtailment habits (e.g., BG10 to BG12) exhibited good performance for household electricity

savings. Electricity savings also grew under the BG10 to BG12 scenarios. By maintaining reduced resource usage for a longer time period (i.e., 2, 3 or 4 years), the average annual household electricity consumption would be reduced by 160 kWh/household/year, 234 kWh/household/year and 276 kWh/household/year in 2050, respectively. The reduction in shower frequency (BG4 to BG6) and inclusion of more vegetables in the daily diet (BG7 to BG9) did not play a significant role in household electricity consumption in Naples. In fact, the current share of household electricity consumption for water heating as well as cooking is approximately 15% and 7%, respectively.

## Average annual household gas use

Fig 3 shows the simulated scenarios for average annual gas usage in the metropolitan city of Naples.

According to the computed results, household gas consumption would decrease the most when it was influenced by reducing residents' daily washing lengths. This result reflects the strong interlinkage between the energy and water subsystems in households. Gas use in households in Naples would decrease by 470 to 1,400 kWh/household/year if the daily washing length decreased by 0.8 to 2.4 minutes (BG1 to BG3). Lower water consumption would lead to reduced gas consumption because of the 15% share of gas that is used for water heating in Naples. Preserving the habits of gas savings (BG10 to BG12) is second only to reductions in time (BG1 to BG3). In particular, 620 to 1,070 kWh/household/year could be saved if saving behaviours are maintained for a longer time period, up to 3 years. Currently, only 5% of gas consumption is used for cooking. Therefore, reducing meat and dairy consumption rates (BG7 to BG9) would not significantly impact gas consumption, which would vary from 60 kWh/household/year to 180 kWh/household/year.

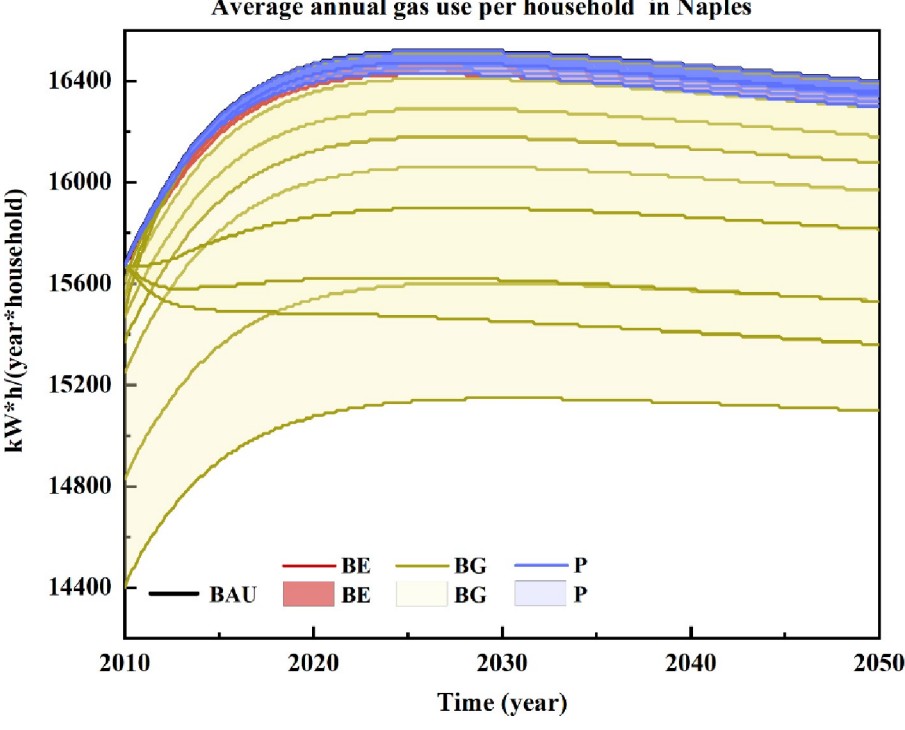

**Fig 3. Scenarios for average annual household gas use in the metropolitan city of Naples for the years 2010–2050.** Included scenarios: Business as usual (BAU), Price-based policies (P), Behavioural education actions (BE), and Behavioural guidance actions (BG).

Price adjustments for all resources considered would equally impact gas consumption. However, direct adjustment of gas prices would exhibit the best performance among the price strategies and lead to savings from 50 to 110 kWh/household/year. The weak effect of price strategies on household resource consumption may depend on household needs for such resources.

## Average annual household water use

Fig 4 shows alternative scenarios for average annual household water use in the metropolitan city of Naples. In this respect, educational actions would not significantly affect household water consumption in Naples, which would vary between 700 and 1,200 L/household/year, with approximately 1/10 of the water savings being related to the reductions in daily washing lengths.

As shown in Fig 4, instructing residents to reduce daily washing lengths (BG1 to BG3) would produce the best performance with respect to household water savings. Even if declining over time, the water-saving effects under the three policy scenarios (BG1 to BG3) are still the highest among all simulated scenarios. In detail, each household in Naples could save 84,100 L/year by reducing its daily washing length to 1.6 minutes each time. By shortening the washing lengths by 0.8 minutes and 1.6 minutes, water savings could also reach 28,000 L/household/year and 56,000 L/household/year, respectively. Reducing shower frequencies (BG4 to BG6) would also bring benefits. Although these benefits would be lower than the BG1 to BG3 scenarios, the water saving potential under scenarios BG4 to BG6 could lead to consumption reductions of 3,200 L/household/year, 6,700 L/household/year and 9,900 L/household/year, respectively.

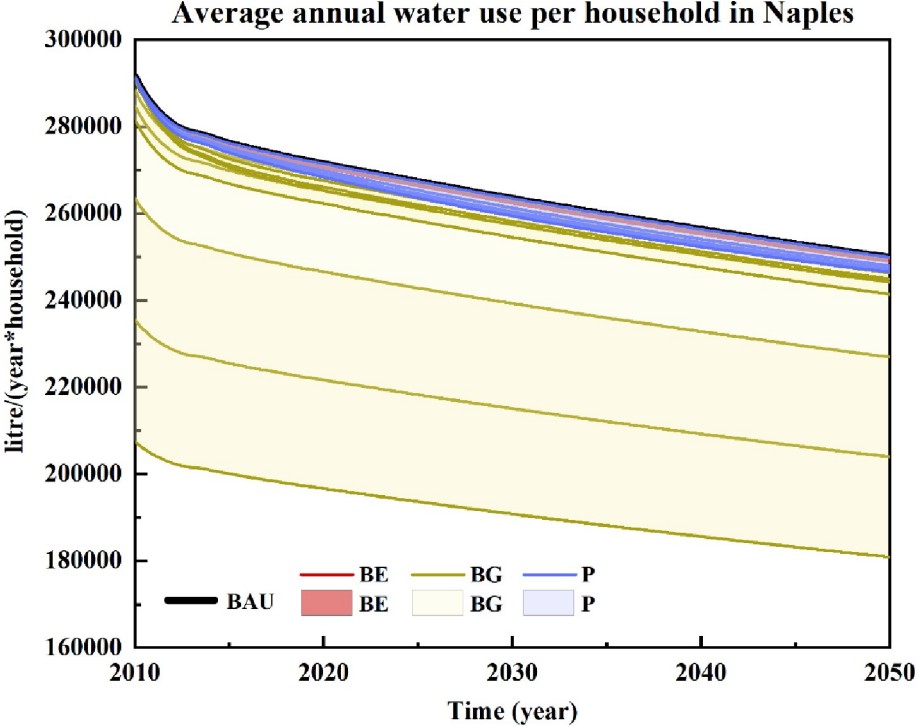

**Fig 4. Scenarios of average annual household water use in the metropolitan city of Naples for the years 2010–2050.** Included scenarios: Business as usual (BAU), Price-based policies (P), Behavioural education actions (BE), and Behavioural guidance actions (BG).

Diet type would not significantly impact household water consumption in Naples. Price-based strategies would perform better in the middle term. Household water savings would be 2,300 L/household/year, 3,200 L/household/year, 3,800 L/household/year and 4,200 L/household/year in 2035 by imposing price growths of 40%, 60%, 80% and 100%, respectively. Policies that would support educational activities would have little effect on reducing water consumption. Consequently, attention should be directed towards policies that support price increases or information campaigns.

### Price policies

Price adjustments have the best regulation effect on natural gas and are followed by water and electricity. However, monetary actions would have little impact on final resource savings. Price growth would trigger consumption costs. This would encourage people to save resources. However, considering the need for these resources, price-oriented policies would affect excess consumption more than encouraging further savings. This fact is proven by the preferences mentioned in the field survey, in which only 13% of respondents favoured adjusting their price adjustment behaviours.

### Behavioural education

Environmental protection policies oriented towards educational actions exhibited the best regulation effect on water. The same trend was not observed in the case of electricity and gas. In particular, the mid-term scenario was better than the long-term scenario. In fact, the impacts of educational campaigns require a longer time from the cultivation of awareness to implementation of specific behaviours. The form and contents of communication and educational campaigns and related policies are also relevant for their success. As shown in the literature, the water-saving effects of environmental publicity activities were lower than those of other policy tools for the water-saving effects of different policies in California [31]. At the same time, the effect of reducing household water consumption could not even be realized through water-saving publicity policies [32]. Generally, implementation of an information publicity policy alone is not an effective strategy.

### Behavioural guidance

Compared with behavioural education, behavioural guidance can provide clearer and more specific information and codes of conduct under basic living requirements. Supporting reductions in time and frequency of resource use and by triggering the preservation of saving behaviours, water consumption could be reduced significantly. The results are confirmed by the literature, which confirms the role of behavioural guidance, which can bring great energy and water conservation benefits [33,34].

## Discussion

The results are first discussed in terms of their sensitivity with respect to initial household preferences. Then, the discussion explores the use of such scenarios to support informed policy actions.

The sensitivity of electricity, gas and water consumption to the differences in household attitudes was tested to determine the relevance of the impacts of different classes of behavioural factors to the final consumption trends. Detailed data of the sensitivity analysis are given as supplementary material (Section S5 in S1 File).

Fig 5 shows the variations in consumption trends with respect to the BAU scenario, which depend on the variations of initial preference values. In particular, a variation between– 20%

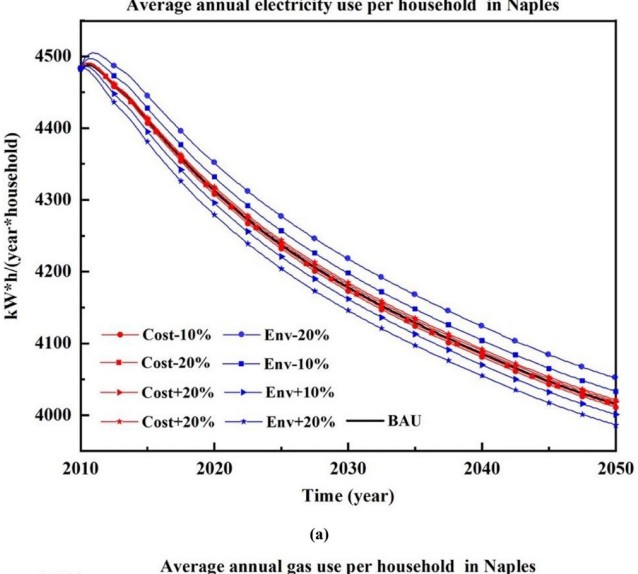

(a)

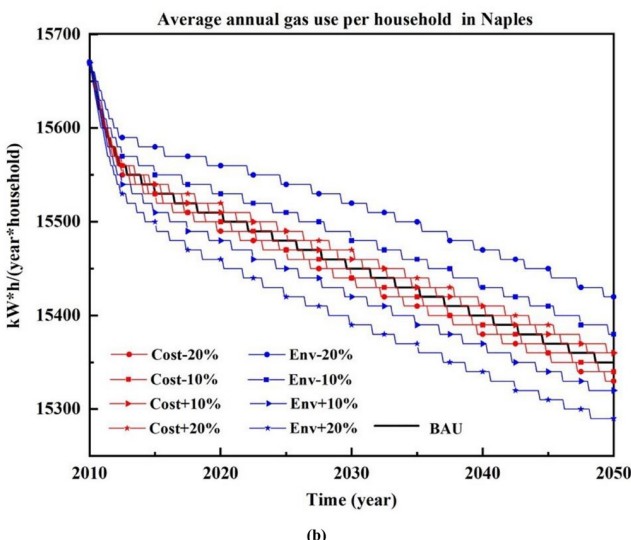

(b)

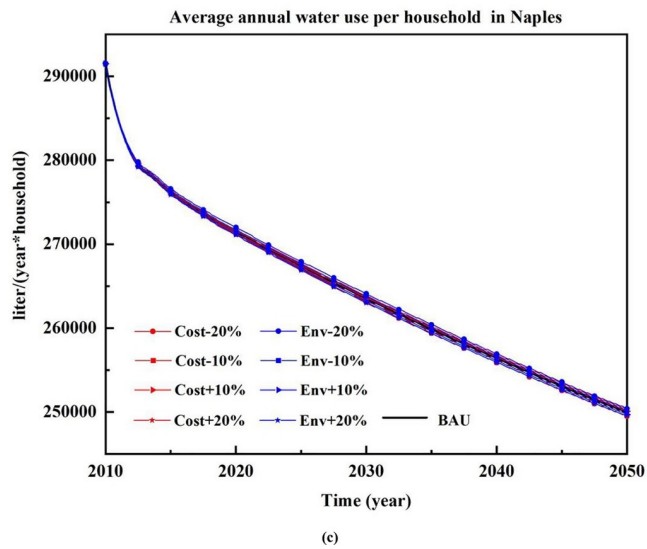

(c)

**Fig 5.** Sensitivity of electricity (a), gas (b) and water (c) use as a function of price and pro-environmental preferences. The percentage variations of the cost-related and behavioural preferences are assessed to observe the variations of consumption trends against the BAU scenario.

and + 20% was imposed on the initial cost of environmentally related attitudes. The factors that impacted the consumption variations more strongly were the environmental factors in relation to pro-environmental behaviours.

The highest impacts were observed for the case of electricity consumption. In fact, a growth of 20% in the attitudes for pro-environmental behaviour of final electricity consumption reduced this consumption with respect to the BAU scenario by 1%. The opposite, i.e., a decline of 20% in pro-environmental behaviour, impacted final electricity consumption with an increase of approximately 1%. In the case of gas consumption, the observed variation with respect to the BAU scenario was ± 0.6%, while in the case of water consumption, it was ± 0.2%. The results indicate that pro-environmental attitudes impact electricity consumption more than gas and water consumption.

The same variations were not observed in the case of market cost variations. In fact, the changes in attitudes within the same range produced impacts that were within ± 0.2% for electricity, gas and water. Thus, the results of the sensitivity tests indicate the relevance of collecting accurate data on pro-environmental attitudes, which have a higher impact on the reliability of the final results.

Another aspect, in addition to the relevance of the data collection phase, which was proved by the sensitivity analysis, is the application of simulations to support policy-making. The relevance of simulation and visualization tools in urban environmental planning is already acknowledged in the literature [35]. Considering the rapid growth of cities and considering the environmental, social and economic impacts of such a phenomenon, the quality of life of citizens may be at risk. The opportunity for rapid data collection, which is supported by the existing and developing information and communication infrastructures, may help urban planners to improve the quality of the modelled scenarios for the future of cities [36].

Simulations can be meaningful instruments to support planning actions by identifying the most appropriate actions, triggers and opportunities under uncertain conditions [37]. Two lines of implementation in this direction could be possible. The first is the integration of simulation results with mapping capabilities by considering the availability of geographical information system (GIS) tools. This process would require collecting more detailed information during the data collection phase [38] while respecting privacy issues according to national and international regulations. Such an option, which was scaled to the household level, was successfully tested in previous studies, as proven by the literature [39–41]. A second line of implementation is the use of simulations to support communication between scientists and policy-makers [42], as well as between policy-makers and citizens [43]. Surely, this option, whose validity is confirmed by the literature, is still underexplored, especially to foster the growth of bottom-up policies and sharing of pro-environmental practices. In particular, supporting transparency in the process of informed policy-making, careful use of resource consumption scenarios and their shared visualization can improve awareness of the impacts of household behaviour on environmental resource conservation.

## Conclusions

This work proposes a set of household-level scenarios for electricity, gas and water consumption that test the efficacy of individual preferences as well as the effectiveness of policies oriented towards pricing refinements as well as educational and communication actions to

improve the conservation of key resources and the adoption of pro-environmental behaviours. The scenarios were developed in the context of the metropolitan city of Naples (southern Italy), which was used as a case study. With respect to the considered case, the results proved a higher efficacy for policies that were oriented towards communication actions to enhance the adoption of saving behaviours, as well as their longer preservation over time. Conversely, monetary policies did not impact households to the same extent, considering that electricity, gas and water are basic household resources.

Implementation of the proposed approach, which considered the mutual relationships among different key resources, could be tested in other urban contexts, which would further implement their use to support the development of longer-term and more effective pro-environmental policies.

## Supporting information

**S1 File.**
(DOCX)

## Acknowledgments

We thank the students of SU who participated in the course of Life Cycle Assessment at the University of Napoli 'Parthenope' and who supported the study with the collection of household data.

## Author Contributions

**Conceptualization:** Marco Casazza, Gengyuan Liu, Sergio Ulgiati.

**Data curation:** Jingyan Xue, Shupan Du.

**Formal analysis:** Shupan Du.

**Funding acquisition:** Gengyuan Liu, Sergio Ulgiati.

**Investigation:** Marco Casazza, Jingyan Xue, Shupan Du.

**Methodology:** Marco Casazza, Jingyan Xue, Shupan Du.

**Project administration:** Gengyuan Liu, Sergio Ulgiati.

**Software:** Jingyan Xue, Shupan Du.

**Supervision:** Marco Casazza, Gengyuan Liu, Sergio Ulgiati.

**Validation:** Jingyan Xue, Shupan Du.

**Visualization:** Jingyan Xue, Shupan Du.

**Writing – original draft:** Marco Casazza, Jingyan Xue, Shupan Du, Gengyuan Liu, Sergio Ulgiati.

**Writing – review & editing:** Marco Casazza, Jingyan Xue, Shupan Du, Gengyuan Liu, Sergio Ulgiati.

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
