## [Decision Letter · Decision Letter 0]

11 Feb 2021

PONE-D-20-40863

Simulation of scenarios for urban household water and energy consumption

PLOS ONE

Dear Dr. Casazza,

Thank you for submitting your manuscript to PLOS ONE. After careful consideration, we feel that it has merit but does not fully meet PLOS ONE’s publication criteria as it currently stands. Therefore, we invite you to submit a revised version of the manuscript that addresses the points raised during the review process.

We look forward to receiving your revised manuscript.

Kind regards,

Ghaffar Ali, PhD

Academic Editor

PLOS ONE

Journal Requirements:

2) We suggest you thoroughly copyedit your manuscript for language usage, spelling, and grammar. If you do not know anyone who can help you do this, you may wish to consider employing a professional scientific editing service.  

3) Please include captions for your Supporting Information files at the end of your manuscript, and update any in-text citations to match accordingly. Please see our Supporting Information guidelines for more information: http://journals.plos.org/plosone/s/supporting-information.

Reviewers' comments:

Reviewer's Responses to Questions

**Comments to the Author**

1. Is the manuscript technically sound, and do the data support the conclusions?

Reviewer #1: Yes

Reviewer #2: Yes

2. Has the statistical analysis been performed appropriately and rigorously? 

Reviewer #1: Yes

Reviewer #2: Yes

3. Have the authors made all data underlying the findings in their manuscript fully available?

Reviewer #1: Yes

Reviewer #2: Yes

4. Is the manuscript presented in an intelligible fashion and written in standard English?

Reviewer #1: Yes

Reviewer #2: Yes

5. Review Comments to the Author

Reviewer #1: This article simulations of scenarios arising from varying degrees of household water and energy consumption in the Napoli metropolitan city of Italy. The findings arising from the research are important and timely. The article is very structured and reads well. It can be considered for publication in the PLOS ONE with minor revisions.

Introduction: It is good, but it lacks a comprehensive setup. Authors could consider revising it as it is not linked up nicely especially the first paragraph seems more like in bits and pieces.

Methods and Results

Technically strong and well explained. However, I could have condensed the methods with an explanation given in the supplementary text (Just a suggestion).

Discussion and Conclusion

Very sound and well written. No comments.

Reviewer #2: See comments in attachment.

To the best of my knowledge the manuscript is technically sound and seriously support the conclusion, official statistical data was obtained by the authors and a sample survey was carried out using 248 university students as volunteers and the data was made available. Statistical analysis were performed. Detail of these analysis can be obtained from supplementary sheets provided by authors. Finally the manuscript was written using standard English with some grammatical errors. In general the language was clear and sound and it meet the specification by PLOS ONE JOURNAL.

6. PLOS authors have the option to publish the peer review history of their article (what does this mean?). If published, this will include your full peer review and any attached files.

Reviewer #1: No

Reviewer #2: No

---

## [Author Response · Author response to Decision Letter 0]

8 Mar 2021

Dear Editor,

Dear Reviewers,

Thank you very much for your efforts and work to support us in improving the quality of our manuscript. 

Introductory reply

We agree with all the indications given by the reviewers. We noticed the presence of several grammar errors. In order to remove them and to improve the quality of the manuscript, we required the support of AJE professional editing services, being one of the companies indicated by PLOS ONE website. In particular, the manuscript was edited for proper English language (British English), grammar, punctuation, spelling, and overall style.

Following the professional editing indication, we modified the title from “Simulation of scenarios for urban household water and energy consumption” to “Simulations of scenarios for urban household water and energy consumption”.

Besides the language revision, the quality editing identified 1 error in Figure 5 (the lack of separation between “Time” and “(year)”), which was corrected. The figure originally appeared to be Figure 7 (before moving Figure 1 and Figure 2 to the Supplementary Materials file, numbering them as Figure 1S and Figure 2S). The words “Non meat & diary” was modified by the editing service as “Nonmeat and nondairy”. Since the same text was reported in Figure 2 (now, Figure 2S), we revised the figure accordingly. 

Following the indications given by Reviewer #1, we moved part of the method, related to details about the simulator structure, in the supplementary materials file, section S1. In fact, we believe that this suggested option enhances the readability of the manuscript. Consequently, we re-numbered the Supplementary Material file sections accordingly. 

Following the indications given by Reviewer #2, we improved the brightness of the legends in Figures 4, 5 and 6 (now, Figures 2, 3 and 4). Besides the sub-section titles size (originally 12 pt, now 11 pt), the style review by AJE detected some inconsistencies in the main text (sometimes, the main text was 12 pt, while, in other parts, the text size was 11 pt). We modified the main text size, using a uniform 11 pt size.

Figure 1 and Figure 2 are now numbered as Figure 1S and Figure 2S, since they are now found in the Supplementary Materials file. Consequently, the numbering of figures has changed. Figures 4, 5 and 6 are now Figures 2, 3 and 4. All figures numbers have been changed and checked. The same check has been performed along the text to have the correct relation between the main text and figures. 

Detailed response to reviewers

Reviewer #1: 

1. Introduction: It is good, but it lacks a comprehensive setup. Authors could consider revising it as it is not linked up nicely especially the first paragraph seems more like in bits and pieces.

Thank you for your comment. The poor language quality made the text not linked up nicely, as you state here. This is why we asked a language revision, also including a style revision. 

2. Methods and Results. Technically strong and well explained. However, I could have condensed the methods with an explanation given in the supplementary text (Just a suggestion).

Thank you for your comment. Thinking about you suggestion, we decided to move the sub-section “Household FEW nexus structure” to the supplementary materials file, section S1. In fact, we believe that this suggested option enhances the readability of the manuscript. Consequently, we added a sentence in the main text to indicate the presence of section S1. We also re-numbered the following supplementary materials sections accordingly. 

Figure 1 and Figure 2 are now numbered as Figure 1S and Figure 2S, since they are now found in the Supplementary Materials file. Consequently, the figures numbering has changed. Figures 4, 5 and 6 are now Figures 2, 3 and 4.

Reviewer #2

1. There are a lot of grammatical errors in the manuscript

We agree with your comment. In order to remove them and to improve the quality of the manuscript, we required the support of AJE professional editing services, being one of the companies indicated by PLOS ONE website. In particular, the manuscript was edited for proper English language (British English), grammar, punctuation, spelling, and overall style.

2. The legends in Figures 4, 5 and 6 should be made brighter

We agree with your comment. We modified the legends of the three figures accordingly.

Please, note that Figure 1 and Figure 2 are now numbered as Figure 1S and Figure 2S, since they are now found in the Supplementary Materials file. Consequently, the numbering of figures has changed. In particular, Figures 4, 5 and 6 are now Figures 2, 3 and 4. All figures numbers have been changed and checked. The same check has been performed along the text to have the correct relation between the main text and figures. 

3. In organization of paper, authors only bolded Sections, I suggest that the subsections should also be bolded but different font size.

We agree with your comment. We noticed the problem. We modified the text in the following way:

Section title: Times New Roman, bold, 12 pt

Sub-section title: Times New Roman, bold, 11 pt

Main text: Times New Roman, normal, 11 pt

Figures and tables captions: Times New Roman, normal, 10 pt

Final remarks

We include the professional revision certificate together with the reviews files.

Finally, we hope that you will appreciate our efforts in improving the manuscript quality, following your requirements and suggestions.

---

## [Decision Letter · Decision Letter 1]

25 Mar 2021

Simulations of scenarios for urban household water and energy consumption

PONE-D-20-40863R1

Dear Dr. Casazza,

We’re pleased to inform you that your manuscript has been judged scientifically suitable for publication and will be formally accepted for publication once it meets all outstanding technical requirements.

Kind regards,

Ghaffar Ali, PhD

Academic Editor

PLOS ONE

Additional Editor Comments (optional):

Reviewers' comments:

Reviewer's Responses to Questions

**Comments to the Author**

1. If the authors have adequately addressed your comments raised in a previous round of review and you feel that this manuscript is now acceptable for publication, you may indicate that here to bypass the “Comments to the Author” section, enter your conflict of interest statement in the “Confidential to Editor” section, and submit your "Accept" recommendation.

Reviewer #1: All comments have been addressed

Reviewer #2: All comments have been addressed

2. Is the manuscript technically sound, and do the data support the conclusions?

Reviewer #1: Yes

Reviewer #2: (No Response)

3. Has the statistical analysis been performed appropriately and rigorously? 

Reviewer #1: Yes

Reviewer #2: (No Response)

4. Have the authors made all data underlying the findings in their manuscript fully available?

Reviewer #1: No

Reviewer #2: (No Response)

5. Is the manuscript presented in an intelligible fashion and written in standard English?

Reviewer #1: Yes

Reviewer #2: (No Response)

6. Review Comments to the Author

Reviewer #1: Authors have addressed all the comments in the revised version. I am now happy to accept the manuscript.

Reviewer #2: The current version of the manuscript is a significant improvement over the previous one. All the concerns I raised in the previous manuscript have been addressed by the authors. Therefore, without any reservation, the manuscript should be considered for publication.

7. PLOS authors have the option to publish the peer review history of their article (what does this mean?). If published, this will include your full peer review and any attached files.

Reviewer #1: No

Reviewer #2: No

---

## [Editor Report · Acceptance letter]

26 Mar 2021

PONE-D-20-40863R1 

Simulations of scenarios for urban household water and energy consumption 

Dear Dr. Casazza:

I'm pleased to inform you that your manuscript has been deemed suitable for publication in PLOS ONE. Congratulations! Your manuscript is now with our production department. 

Kind regards, 

on behalf of

Dr. Ghaffar Ali 

Academic Editor

PLOS ONE